# Identification and Analysis of the Expression of the *PIP5K* Gene Family in Tomatoes

**DOI:** 10.3390/ijms25010159

**Published:** 2023-12-21

**Authors:** Zepeng Wang, Zhongyu Wang, Xianguo Li, Zhaolong Chen, Yuxiang Liu, Fulin Zhang, Qi Dai, Qinghui Yu, Ning Li

**Affiliations:** 1Key Laboratory of Genome Research and Genetic Improvement of Xinjiang Characteristic Fruits and Vegetables, Institute of Horticultural Crops, Xinjiang Academy of Agricultural Sciences, Urumqi 830091, China; zx1119802932@163.com (Z.W.); wzy_caas@163.com (Z.W.); 15099198840@163.com (X.L.); 17866711897@163.com (Z.C.); lyx1364199850@163.com (Y.L.); 18095998056@163.com (F.Z.); daiqixj@163.com (Q.D.); 2The State Key Laboratory of Genetic Improvement and Germplasm Innovation of Crop Resistance in Arid Desert Regions (Preparation), Urumqi 830091, China; 3College of Horticulture, Xinjiang Agricultural University, Urumqi 830052, China

**Keywords:** tomato, *PIP5K* gene family, fruit development, salt stress, expression analysis

## Abstract

To explore the function of phosphatidylinositol 4-phosphate 5-kinase (PIP5K) in tomatoes, members of the tomato PIP5K family were identified and characterized using bioinformatic methods, and their expression patterns were also analyzed under salt stress and in different tissues. Twenty-one PIP5K members—namely, *SlPIP5K1*–*SlPIP5K21*—were identified from ten chromosomes, and these were divided into three groups according to a phylogenetic analysis. Further bioinformatic analysis showed four pairs of collinear relationships and fragment replication events among the *SlPIP5K* family members. To understand the possible roles of the *SlPIP5K*s, a cis-acting element analysis was conducted, which indicated that tomato *PIP5K*s could be associated with plant growth, hormones, and stress responses. We further validated the results of the in silico analysis by integrating RNA-seq and qRT-PCR techniques for salt- and hormone-treated tomato plants. Our results showed that *SlPIP5K* genes exhibited tissue- and treatment-specific patterns, and some of the *SlPIP5K*s exhibited significantly altered expressions after our treatments, suggesting that they might be involved in these stresses. We selected one of the *SlPIP5K*s that responded to our treatments, *SlPIP5K2*, to further understand its subcellular localization. Our results showed that *SlPIP5K2* was located on the membrane. This study lays a foundation for the analysis of the biological functions of the tomato *SlPIP5K* genes and can also provide a theoretical basis for the selection and breeding of new tomato varieties and germplasm innovation, especially under salt stress.

## 1. Introduction

Phosphatidylinositol (PI) and its derivatives have important signaling functions in plants. These molecules transmit signals from the external environment to plant cells, leading to various physiological and biochemical responses that help the plant adapt to its surroundings [1,2]. Phosphatidylinositol 4-phosphate 5-kinase (PIP5K) is a key enzyme in the PI signaling pathway, and it regulates a wide range of cellular functions [3,4]. Plant PIP5K proteins typically have multiple MORNs at the *N* terminus and a conserved catalytic PIPKc domain at the C terminus. MORNs are unique to the plant PIP5K family, and the PIPKc domain is highly conserved in eukaryotes [5,6]. It has been shown that the *PIP5K* genes are widespread in organisms [7,8] and have multiple isoenzymes that are encoded by a family of genes [9,10,11]. There are three PIP5K isozymes in mammals, and these are involved in different physiological processes, such as cytoskeletal dynamics, cell division, and apoptosis [12,13,14]. Similarly, the *PIP5K* genes are essential in fungi, where they are involved in vesicle function and morphology [15]. These results suggest that the *PIP5K* genes play a crucial role in both animals and fungi.

In plants, *PIP5K* genes are responsible for encoding phosphatidylinositol isoenzymes, an important gene family that regulates the development of several tissues and organs in plants. For example, *AtPIP5K3* regulates the apical elongation of root hairs, *AtPIP5K1* and *AtPIP5K2* promote pollen development [16,17], and *AtPIP5K4* and *AtPIP5K5* are involved in pollen germination and pollen tube growth [18]. In addition, *AtPIP5K1* has been implicated in the response to water stress, as well as in the ABA signaling pathway [19]. In monocot plants like rice and maize, *OsPIP5K1* plays a key role in rice tasseling, and *ZmPIP5K* genes can promote root hair growth [7,20].

Thorough investigations of the PIP5K family members across various plant species, including Arabidopsis (*Arabidopsis thaliana*), rice (*Oryza sativa*), ginkgo biloba (*Ginkgo biloba*), soybean (*Glycine max*), wheat (*Triticum aestivum*), and watermelon (*Citrullus lanatus*), have been conducted [2,4,6]. However, limited information is available regarding the characterization of this family in tomato plants. Tomato (*Solanum lycopersicum*) is a perennial herbaceous plant of the Solanaceae family, rich in a variety of vitamins, carotenoids, lycopene, minerals, and other ingredients that can lower blood pressure and cool people down, and remove toxins, thus playing a very important role in human health. Tomato plants, in many parts of the world, are cultivated in large quantities, and China’s tomato production accounts for about a quarter of the world’s tomato production. With the level of consumption continuously increasing, people’s dietary habits have gradually changed, causing an increase in the demand for healthy foods, including tomatoes. Ultra-processed foods and healthcare products tend to be people’s favorites. China accounts for a third of the world’s trade in tomato paste, while tomato planting, processing, and exports have sustained an upward trend. But processed tomatoes can only be grown in a limited number of suitable regions, with prime production areas located in Europe, California, Xinjiang, China, and the Inner Mongolian Loop. Therefore, the discovery of the potential function of genes within the tomato will provide important theoretical support for increasing tomato yields and enhancing the quality of tomatoes produced, as well as improving their resistance to adversity and stress [21,22,23]. This study used bioinformatic methods to identify the tomato *PIP5K* gene family and study the structure, conserved structural domains, and tissue-specific expression of these genes. We also analyzed the properties of the tomato PIP5K family members in terms of their phylogeny, chromosomal distribution, protein physicochemical properties, and motif prediction. Meanwhile, the promoter cis-acting elements, expression patterns, and protein interaction networks of the family member genes were analyzed, and RNA-seq and qRT-PCR were used to analyze the expression patterns of the tomato *SlPIP5K* genes under salt stress. The results can provide a theoretical basis for further investigation of the regulatory mechanism and function of PIP5K in tomatoes.

## 2. Results

### 2.1. Characterization of Members of the SlPIP5K Gene Family in Tomato

In this study, 21 putative *SlPIP5K* genes were initially identified in the tomato plant using genome-wide identification analysis, and *SlPIP5K1*–*SlPIP5K21* were named based on the position of the family members in the genome chromosome. The results of protein physicochemical property analysis showed (Table 1) that the amino acid residues in *SlPIP5K* family members ranged from 111 aa to 1921 aa, and the protein molecular weights ranged from 12,389.00 aa to 214,107.32 aa, of which *SlPIP5K16* was the smallest at 111 aa and 12,389.00 aa, respectively; *SlPIP5K3* was the largest at 1921 aa and 214,107.32 aa, respectively. The isoelectric points ranged from 4.80 to 9.12, with *SlPIP5K18* being the lowest and *SlPIP5K12* being the highest. The *SlPIP5K* gene family proteins all exhibit hydrophilicity.

The secondary structures of proteins encoded by the *SlPIP5K* gene family are mainly α-helical and β-folded with irregular coiling. The 21 proteins encoded by the *SlPIP5K* gene family are predominantly α-helical (from 22.96% to 35.79%) and irregularly coiled (from 34.92% to 52.69%), with less β-folding (from 3.41% to 11.40%).

### 2.2. Chromosomal Localization of the SlPIP5K Gene Family in Tomato

The chromosomal localization analysis results showed that 21 tomato *SlPIP5K* gene family members were distributed on 10 chromosomes (Figure 1), of which the *SlPIP5K* genes were mainly distributed on Chr1 (2), Chr4 (2), Chr10 (3), Chr11 (2), and Chr12 (7), and there were no *SlPIP5K* gene family members on Chr2 and Chr8. The remaining five chromosomes each have one *PIP5K* member. The distribution of tomato *SlPIP5K* gene family members on chromosomes is heterogeneous; among them, the tomato *SlPIP5K* genes were distributed at the upper end of the chromosomes Chr4, Chr5, Chr7, Chr9, and Chr11 and the lower end of the chromosomes Chr1, Chr3, Chr6, and Chr10, whereas the members of the gene family *SlPIP5K15*–*SlPIP5K20* were more centrally distributed at the upper end of the chromosome Chr12, and *SlPIP5K21* was distributed at the lower end.

### 2.3. Phylogenetic Classification of the SlPIP5K Gene in Tomato

To clarify the phylogenetic relationship of *PIP5K* genes, a phylogenetic tree was constructed for the protein sequences of a total of 58 PIP5K conserved structural domains from *Arabidopsis thaliana* (*AtPIP5K*, 15), potato (*StPIP5K*, 12), grapevine (*VvPIP5K*, 10), and tomato (*SlPIP5K*, 21) (Figure 2). Based on the protein sequences of the *PIP5K* family members, a phylogenetic tree was constructed, and the *PIP5K* genes from the four species were classified into three groups, I, II, and III, and the *PIP5K* gene family members in the same subgroup were more closely related to each other. Among them, group I has a total of thirty-one *PIP5Ks*, containing thirteen *SlPIP5Ks*, eight *AtPIP5Ks*, five *StPIP5Ks,* and five *VvPIP5Ks*; there were a total of twelve *PIP5K* genes in group II, containing two *SlPIP5K*, five *AtPIP5K*, three *StPIP5K,* and two *VvPIP5K*; and group III had a total of fifteen *PIP5K* genes, containing six *SlPIP5K*, two *AtPIP5K*, four *StPIP5K,* and three *VvPIP5K*. Group I was divided into two subgroups, Ia and Ib, containing six and seven *SlPIP5K* gene family members, respectively; group III was divided into two subgroups, IIIa and IIIb, and only group IIIb contained six *SlPIP5K* genes. The genes within the same group usually have similar functional characteristics, and the majority of the tomato *SlPIP5K* genes are in groups I and III, which suggests that the *SlPIP5K* genes mainly carry out this function in tomato plants.

### 2.4. Tomato SlPIP5K Gene Family Member Motifs and Gene Structures

As shown in Figure 3, PIP5K proteins have different motif types. Among them, *SlPIP5K18*, *SlPIP5K15*, *SlPIP5K1*7, *SlPIP5K19,* and *SlPIP5K16* contain only coding DNA sequence (CDS), while the other genes have both CDS and untranslated region (UTR). Gene structure analysis revealed that the number of introns in the tomato *SlPIP5K* genome ranged from one to thirteen, with *SlPIP5K16* containing only one intron, *SlPIP5K3* and *SlPIP5K12* containing thirteen introns, and the rest of the *SlPIP5K* gene family members contained from two to twelve introns. Introns are sequences that block the linear expression of genes, which increases the length of genes, increases the frequency of recombination between genes, favors species evolution, and has regulatory effects. The results suggest that the *SlPIP5K16* gene may have a function but not a significant role in tomato evolution; presumably, *SlPIP5K3* or *SlPIP5K12* may be more favorable for tomato species evolution. Combined with evolutionary tree analysis, it can be seen that the members of gene families within the same group are more similarly related. For example, *SlPIP5K5*, *SlPIP5K10,* and *SlPIP5K11* are within the same group as *SlPIP5K20*, which are similarly related, and as shown by the conserved motifs, these four *SlPIP5K* genes have similar gene structures, and the motifs of the cohort members are highly similar. It can be seen that the members within the same group have closer clustering relationships and similar gene structures, indicating similar functions. In addition, 21 tomato PIP5K proteins were searched for ten motifs, with three subfamilies of 13 PIP5K proteins sharing motif 1 and motif 8, which indicates that motif 1 and motif 8 are more widely distributed and strongly conserved in the tomato PIP5K protein sequence. Most *SlPIP5K* members in the same subfamily also share some conserved motifs, suggesting that these proteins are functionally conserved. *SlPIP5K16* contains only motif 3, suggesting that *SlPIP5K16* has a particular function. At the same time, some motifs are specifically present in certain subfamilies, suggesting that *SlPIP5K* is functionally diverse in different subfamilies.

### 2.5. Analysis of Protein Structural Domains of Tomato SlPIP5K Gene Family Members

The distribution of the structural domains of SlPIP5K proteins is shown in Figure 4, and the results indicate that all the *SlPIP5K* members contain the PIP5K structural domain. Among them, nine *SlPIP5K* members all contained from five to seven MORN structural domains at the *N*-terminus, and the MORN structural domains were identified for the first time in mammalian articulation as plasma membrane binding modules. Later, in Arabidopsis and rice, MORN structural domains were found to be involved in plasma membrane localization and phosphatidic acid (PA) binding [24,25,26]. Therefore, this function may also exist in tomatoes, but further research is needed. Among these twenty-one *SlPIP5Ks*, seven members contain the Cpn60_TCP1 domain, three members (*SlPIP5K3*, *SlPIP5K8*, and *SlPIP5K21*) contain the FYVE domain, and five members (*SlPIP5K15*, *SlPIP5K16*, *SlPIP5K17*, *SlPIP5K18*, and *SlPIP5K19*) contain only the PIP5K structural domain. In addition, *SlPIP5K12* contains multiple types of structural domains at the C-terminus, suggesting that it may have multiple simultaneous functional expressions in tomato.

### 2.6. Analysis of Covariance and Evolutionary Pressure Analysis for Members of the SlPIP5K Gene Family in Tomato

In this study, we constructed a covariance map between the tomato *SlPIP5K* genes (Figure 5) and analyzed intraspecies covariance for 21 *SlPIP5K* gene family members. The results showed that there were four homologous pairs of tomato *SlPIP5K* genes with covariance between their members *SlPIP5K10* and *SlPIP5K11*, *SlPIP5K14* and *SlPIP5K7*, *SlPIP5K21* and *SlPIP5K3*, and *SlPIP5K4* and *SlPIP5K6*, all of which undergo fragment replication. From the above, it can be seen that gene duplication exists in the *SlPIP5K* gene family, suggesting that the *PIP5K* gene may have undergone family member amplification through duplication during evolution.

To further explore the evolutionary relationships among members of the *PIP5K* gene family, a comparative covariance map of the Arabidopsis *AtPIP5K*, potato *StPIP5K*, grape *VvPIP5K,* and tomato *SlPIP5K* families was constructed in this study (Figure 6). The results showed that straight homologous *PIP5K* gene pairs existed in *Arabidopsis thaliana*, potato, grape, and tomato, with the number of homologous gene pairs between tomato and potato being greater than those between tomato and *Arabidopsis thaliana* and tomato and grape. This suggests that the *PIP5K* gene family of tomato and potato have a closer homologous evolutionary relationship.

In this study, we calculated the Ka/Ks ratio between the replication of the *SlPIP5K* genes and the collinear *PIP5K* genes in tomato to probe the evolution of the *SlPIP5K* gene family and *PIP5K* genes among species. The results (Table 2) showed that all four *SlPIP5K* genes had a Ka/Ks ratio of less than one; thus, it is clear that the members of the *SlPIP5K* gene family underwent strong purifying selection during evolution.

### 2.7. Analysis of Cis-Acting Elements of the Tomato SlPIP5K Gene Family Promoter

The members of the tomato *SlPIP5K* gene family were analyzed for cis-acting elements, which can be classified according to their functions as light-responsive, stress-responsive elements, phytohormone-responsive, and phytohormone-responsive elements (Figure 7). The number of light-responsive elements (Box 4, G-box, GT1-motif, and TCT-motif) was the highest at 338, with the highest number of G-boxes (115), followed by Box 4 (111), and the *SlPIP5K20* gene, which accounted for the highest number of G-box elements, followed by *SlPIP5K5*; the *SlPIP5K7* gene accounted for the highest number of Box 4 elements, followed by *SlPIP5K5*. The G-box is a cis-acting element involved in the light response, and Box 4 is a part of the DNA module involved in the light response, from which it can be inferred that the main regulatory features of these genes are light-related. Moreover, *SlPIP5K5* possessed the highest number of light-responsive elements, suggesting that the tomato *SlPIP5K5* gene is closely related to the light response. Among the stress response elements, ARE, which determines the binding and anaerobically induced expression of regulatory proteins, had the highest number of response elements, totaling 76; ARE elements were detected in all 21 genes, suggesting that the tomato *SlPIP5K* genes may be closely related to the anaerobic response of tomatoes. The total number of drought-associated MBS elements ranked second, with 34, of which a total of 13 promoters were able to detect the MBS elements. There was a total of 21 low-temperature (LTR) response elements and 23 defense and stress (TC-rich repeats) response elements. In addition, several elements related to plant growth and developmental responses were identified, with the highest number of promoter elements related to healing tissue expression (WUN motif), totaling 27, which suggests that it may induce cell division and differentiation, induce plant organ regeneration, and also play a regulatory role in tomato growth and development. The others include cis-acting elements related to maize alcohol soluble protein metabolism (O2-site), endosperm expression (GCN4-motif), and circadian rhythms.

Three of these cis-acting elements are associated with hormone responses, including the abscisic acid response element (ABRE), the methyl jasmonate response element (CGTCA-motif, TGACG-motif), and the salicylic acid response element (TCA-element). Abscisic acid was effective at controlling tomato growth and improving resistance, with up to a total of 91 response elements. Methyl jasmonate was able to induce a defense response in tomato plants, thus showing strong resistance to the disease, with the second highest number of response elements totaling 76. Salicylic acid can both induce tomato cell differentiation and participate in tomato plant resistance to pathogens, with a minimum total of 29 response elements. This suggests that multiple hormones can be involved in regulating the expression of PIP5K, which together promote tomato growth and improve tomato resistance.

### 2.8. Analysis of the Expression Pattern of the SlPIP5K Gene Family in Tomato

To explore the potential biological functions of the *SlPIP5K* genes in tomato growth and development, the expression of the *SlPIP5K* genes in different tissues, as well as under salt stress, was analyzed using tomato RNA-seq data (Figure 8). As shown in Figure 8A, the expression showed that the levels of 21 *SlPIP5K* genes were generally low in the tomato roots, with *SlPIP5K9* having the highest expression level, whereas the expression levels of the different genes varied greatly in the stems, leaves, and fruits. Among them, *SlPIP5K8* had the highest expression in the stems, leaves, and fruits, and *SlPIP5K18*, *SlPIP5K15*, *SlPIP5K17*, *SlPIP5K19*, *SlPIP5K20*, *SlPIP5K10*, *SlPIP5K11*, *SlPIP5K16*, *SlPIP5K1,* and *SlPIP5K12* had low expression levels. The above results indicate that the tomato *SlPIP5K* genes have significant tissue expression specificity, which also implies that some degree of biological functional differentiation may have occurred among the different *SlPIP5K* members.

Figure 8B shows that the highest expression of *SlPIP5K9* was observed at 0 h and 0.5 h of the salt stress treatment, after which, the expression level decreased; the highest expression level of *SlPIP5K13* was observed at 2 h, 6 h, 12 h, and 24 h of the salt stress treatment. In addition, the duration of adversity stress had different effects on the expression of the *SlPIP5K* gene family, such that the expression of *SlPIP5K7* at 6 h of salt stress did not differ from that at 0 h of salt stress, and its expression was significantly downregulated at 0.5 h, 2 h, and 12 h of the salt stress treatment compared with that at 0 h. However, its expression was significantly upregulated at 24 h of the salt stress treatment compared with that at 0 h. The *SlPIP5K13* gene expression level showed a parabolic trend of increasing, and then decreasing with the salt stress treatment time.

### 2.9. Analysis of the SlPIP5K Protein Interaction Network

To better understand the function of the *PIP5K* genes in tomatoes, this study predicted the interactions between 17 SlPIP5Ks proteins based on the STRING online database (Figure 9). Several proteins exhibit direct interactions, such as SlPIP5K16 and SlPIP5K6. In addition, SlPIP5Ks interact with the other regulatory transcription factor proteins, such as Atg6, which may be related to their expression patterns in particular cases. Atg6 proteins have been shown to be involved in the regulation of crop growth and development and in improving stress resistance [27], suggesting that the SlPIP5K protein may be involved in the regulation of the stress response via interactions with Atg6. There is also a portion of already annotated proteins in the interaction network diagram, e.g., Solyc09g015350.2.1 is annotated as phosphatidylinositol 3-kinase, which belongs to the PI3/PI4 kinase family and is involved in hormone signaling [28]; this provides a hint of the possible involvement of SlPIP5Ks in hormone signaling. Solyc04g012170.2.1 is annotated for the WD-40 repeat family proteins, a group of proteins with multiple functions in a wide range of biological processes, such as cell signaling, protein transport, cell division, light signaling, and the formation of meristematic tissues [29], suggesting that the SlPIP5K proteins interact with these proteins to participate in the regulation of tomato growth and development. There are also some proteins that lack annotations. They have clear direct or indirect synergistic interactions with the SlPIP5K proteins, but their function is currently unknown.

### 2.10. Quantitative Fluorescence Analysis of the SlPIP5K Gene in Tomato

Salt stress is a very important environmental factor restricting agricultural production throughout the world, and the identification of salt stress-related genes for breeding stress-tolerant varieties is an urgent problem. To understand the expression level of the *SlPIP5K* genes in tomato under salt and fruit growth hormone treatments, in this study, we examined the transcript levels of *SlPIP5K* genes and their downstream genes in various organ materials of tomato using qRT-PCR and verified the expression patterns of *SlPIP5K* genes in tomato.

An analysis of the expression of the tomato *PIP5K* family in different tissues and organs (Figure 10) showed that the *PIP5K* family members have tissue specific expression patterns. Many of the genes (*SlPIP5K2*, *SlPIP5K3*, *SlPIP5K4*, *SlPIP5K5*, *SlPIP5K6*, *SlPIP5K7*, *SlPIP5K9*, *SlPIP5K15*, *SlPIP5K17*, and *SlPIP5K18*) were highly expressed in the roots; among them, *SlPIP5K1* had the highest expression level of 1.01-fold, and *SlPIP5K21* had the lowest, with no significant difference between them. Some of the genes (*SlPIP5K1*, *SlPIP5K11*, *SlPIP5K13*, *SlPIP5K19*, and *SlPIP5K21*) were highly expressed in the leaves, of which, *SlPIP5K1* had the highest expression level of 36.46-fold, *SlPIP5K4* had the lowest, and the rest of them ranged from 4.43- to 34.44-fold; only *SlPIP5K8* had a high expression level of 65.48-fold in the stems, *SlPIP5K9* was the lowest, and the rest ranged from 7.05- to 57.54-fold, suggesting that the tomato *PIP5K* family genes may be involved in the growth regulation of tomato, but the growth regulation of tomato leaves and stems was not obvious for *SlPIP5K4* and *SlPIP5K9*, respectively.

The expression levels of the *SlPIP5K* genes were determined under NaCl stress at 0 h, 0.5 h, 2 h, 4 h, 8 h, and 12 h (Figure 11). The results showed that the expression of some of the genes (*SlPIP5K2*, *SlPIP5K4*, *SlPIP5K5*, *SlPIP5K8*, *SlPIP5K15,* and *SlPIP5K18*) were upregulated in the treatments compared to those in the control, with higher expression levels of *SlPIP5K4*, *SlPIP5K5*, *SlPIP5K15,* and *SlPIP5K18* after 0.5 h of the NaCl treatment, and the highest expression level of *SlPIP5K18* was 5.50-fold in the 0 h treatment. At 2 h, the expression levels of *SlPIP5K5*, *SlPIP5K8*, *SlPIP5K15,* and *SlPIP5K18* were higher, with the highest expression level of *SlPIP5K5* being 5.82-fold in the 0 h treatment. At 4 h, the expression levels of *SlPIP5K4*, *SlPIP5K5*, *SlPIP5K8*, *SlPIP5K15,* and *SlPIP5K18* were higher, with the highest expression level of *SlPIP5K5* being 5.49-fold in the 0 h treatment. At 8 h, the expression levels of *SlPIP5K4*, *SlPIP5K5*, *SlPIP5K8*, *SlPIP5K15,* and *SlPIP5K18* were higher, with the highest expression level of *SlPIP5K18* being 9.57-fold in the 0 h treatment. At 12 h, the expression levels of *SlPIP5K2*, *SlPIP5K4*, *SlPIP5K5*, *SlPIP5K8*, *SlPIP5K15,* and *SlPIP5K18* were higher, with the *SlPIP5K8* expression level being the highest at 6.50-fold in the 0 h treatment. In addition, *SlPIP5K5*, *SlPIP5K15,* and *SlPIP5K18* were significantly upregulated after the NaCl treatment, suggesting that they may play a greater regulatory role in tomato under salt stress. Among them, their expression levels at 8 h were all the highest at 8.98, 7.39, and 9.57 times more than that of the control, respectively, indicating that these three genes were significantly regulated at 8 h of the salt stress treatment in the tomato. The expressions of *SlPIP5K2* and *SlPIP5K4* were both upregulated after the salt stress treatment, in which, *SlPIP5K2* had the highest expression level of 2.65-fold at 12 h of the salt stress treatment, *SlPIP5K4* had the highest expression level of 2.44-fold at 4 h, but the expression level was low, suggesting that it played a minor regulatory role in the tomato salt stress treatment. The expression levels of some genes (*SlPIP5K1*, *SlPIP5K9*, and *SlPIP5K13*) were decreased, and their expression was repressed; among them, the expression of *SlPIP5K1* showed a gradual decrease during the stress treatment, thus indicating that it negatively regulated the tomato.

Tomato fruit growth was affected by naphthaleneacetic acid (NAA), 2,4-epibrassinolide (EBR), and melatonin (MT), with more significant differences in *SlPIP5K* gene expression between treatments (Figure 12 and Appendix A). At the stage of fruit growth and expansion, the highest expression of *SlPIP5K19* was 4.44 times higher than that of CK induced by 10 mg·L^−1^ NAA hormone; at the green ripening stage of fruit growth, the highest expression of *SlPIP5K19* was 11.95-fold of that of CK induced by 30 mg·L^−1^ NAA hormone, indicating that it positively regulated the growth of tomato fruits and promoted fruit growth and development. The 20 mg·L^−1^ NAA hormone significantly promoted elevated *SlPIP5K* gene expression during the fruit growth and expansion stage; during the green ripening stage of fruit growth, the overall effect of 30 mg·L^−1^ NAA on the enhancement of *SlPIP5K* gene expression was significant, thus indicating that the effect of 30 mg·L^−1^ NAA in regulating the growth and development of tomato fruits was more obvious during this period.

As seen from the figure, 0.1 mg·L^−1^ EBR hormone and 150 µmol·L^−1^ MT hormone significantly promoted the elevated expression of the *SlPIP5K* genes during the green ripening stage of fruit growth. Among them, under the induction of 0.1 mg·L^−1^ EBR hormone, the highest expression level of *SlPIP5K19* was 22.93-fold higher than that of CK, and the higher expression levels of *SlPIP5K2* and *SlPIP5K21* were 11.11-fold and 6.45-fold higher than that of CK, respectively, suggesting that *SlPIP5K2*, *SlPIP5K21,* and *SlPIP5K19* may have a positive regulatory effect on the growth of tomato fruits and improve the fruits’ resistance. Moreover, the *SlPIP5K19* expression level still reached a maximum of 10.00-fold of CK after 150 µmol·L^−1^ MT hormone induction, and the *SlPIP5K2* and *SlPIP5K8* expression levels were significantly elevated. The results suggest that the *SlPIP5K19* gene may have a significant role in the regulation of tomato fruit growth and development and may be involved in the regulation of the stress response.

### 2.11. Analysis of the Subcellular Localization of SlPIP5K2 in Tomato

The expression of *SlPIP5K2* was strongly induced by the salt stress and hormone treatments, suggesting its potential role under these stresses. To further understand *SlPIP5K2*, a study of the transient expression of CAM-EGFP-SlPIP5K2 Agrobacterium in tobacco was conducted. The results showed that intense fluorescence occurred only at the cell membrane, confirming that *SlPIP5K2* is localized at the cell membrane (Figure 13). This suggests that it is involved in tomato growth regulation as a membrane protein.

## 3. Discussion

Phosphatidylinositol 4-phosphate 5-kinase (PIP5K) is a key enzyme in the phosphatidylinositol signaling pathway that plays an important role in plant growth, development, and biotic and abiotic stress responses. Currently, the *PIP5K* gene family has been studied in a variety of plants, including *Arabidopsis thaliana*, rice, Ginkgo biloba, soybean, wheat, and watermelon, and eleven, eleven, twenty-two, twelve, sixty-four, and eight members of the *PIP5K* family have been identified, respectively [2,4,6]; however, little has been reported on the tomato *PIP5K* gene family. The genome-wide analysis of *PIP5K* in tomatoes has contributed to a better understanding of this gene family. In this study, 21 *SlPIP5K* genes were identified from the tomato genome, and gene structure analysis revealed that the number of introns ranged from one to thirteen, which was higher than those found in the crops such as wheat [2] and watermelon [4]. It is hypothesized that the frequency of recombination between the tomato *SlPIP5K* genes may be increased and that this change is more favorable for tomato species’ evolution and has a corresponding regulatory role. Subcellular localization is the determination of where biomolecules carry out their biological functions and whether they are able to do so in the correct way [30]; based on the high expression of *SlPIP5K2* in salt tolerance and fruit development, this study found that *SlPIP5K2* is localized in the cell membrane using confocal microscopy. We hypothesis *SlPIP5K2* functions in the plasma membrane under salt stress.

The phylogenetic and evolutionary relationships of the *PIP5K* genes of four representative species (tomato, *Arabidopsis thaliana*, potato, and grape) were investigated, which showed that the genes were divided into three groups. The *PIP5K* genes within the same group had a high degree of homology, and homologous genes may have come from the same ancestor when duplication occurred and are presumed to have similar functions. Gene duplication is a major factor in the rapid expansion and evolution of gene families [31]. In this study, nine *SlPIP5K* members all contained 5–7 MORN structural domains at the *N*-terminus; similarly, in Arabidopsis, there were nine *AtPIP5Ks* with 7–8 MORN structural domains, five *OsPIP5Ks* with 7–8 MORN structural domains in rice, and six *ClaPIP5K* members with 7–8 MORN structural domains in watermelon [4]. In addition, all the members have the typical structural domain of PIP5K, and the remaining structural domains are only present in the individual genes. This suggests that PIP5K, a key enzyme in the PI signaling system for plant growth and development and response to environmental stress, has been relatively conserved throughout tomato evolution. Studying the Ka/Ks ratio in the replicated genes is an effective way to study their evolutionary impact. The Ka/Ks ratios in the four homologous gene pairs in the *SlPIP5K* members were all less than one, indicating that these gene pairs are subject to strong purifying selection and that the tomato *PIP5K* gene family has stabilized during long-term evolution.

In this study, there were four homologous pairs of tomato *SlPIP5K* genes with covariance between their members. This result further suggests that the *PIP5K* genes have a high degree of homology and that gene duplication exists in the family members, suggesting that the *PIP5K* genes may have undergone family member amplification through duplication during evolution. To further infer the phylogenetic mechanism of the tomato *PIP5K* family, we constructed comparative covariance maps for the four representative species. The results showed that the number of homologous gene pairs between tomato and potato was greater than those between tomato and *Arabidopsis thaliana* and tomato and grape. This suggests that the *PIP5K* gene family of tomato and potato have a closer homologous evolutionary relationship.

Cis-regulatory elements are important molecular switches involved in the transcriptional regulation of genes under abiotic or biotic stress and play an important role in the transcriptional regulation of gene expression [32,33]. In this study, an analysis of the cis-acting elements in the 2000 bp promoter region upstream of *SlPIP5K*s revealed that the promoters of *SlPIP5K*s contain a variety of elements related to plant growth and development as well as cis-acting elements related to abiotic stresses, such as the cis-acting elements involved in cell cycle regulation, hormone-responsive elements, and low-temperature and drought responses. Meanwhile, most of the members in this family respond to a variety of hormone regulations, including ABA, MeJA, and other hormones involved in signal transduction in plant stress, suggesting that the *SlPIP5K* family plays an important role in hormone-regulated plant growth and development. Among them, most of the *SlPIP5Ks* responded to the regulation of MeJA and ABA. Hormones such as ABA and MeJA can alleviate stress damage to plants by increasing the activity of antioxidant enzymes, scavenging free radicals, and increasing the content of osmotically regulated substances [34]. Therefore, it is hypothesized that *SlPIP5Ks* may be involved in the regulatory effects of abiotic stresses by responding to hormonal responses, such as ABA and MeJA, thereby enhancing the plants’ tolerance to adversity stresses, but the specific mechanisms need to be studied in depth.

By analyzing the expression of *SlPIP5K* genes in different tissues, as well as under salt stress, it was clarified that the expression levels of different *SlPIP5K* genes varied greatly in various tomato tissues, which indicates that the tomato *SlPIP5K* genes has significant tissue expression specificity, implying that a certain degree of biological functional differentiation may have occurred between different *SlPIP5K* members. The expression of the *SlPIP5K13* gene was upregulated after the salt stress treatment, which is hypothesized to play a positive regulatory role in the response of tomato to salt stress; the expression levels of the *SlPIP5K5* and *SlPIP5K9* genes were downregulated after the salt stress treatment, which is hypothesized to play a negative regulatory role in the response of tomato to salt stress. We predicted interactions between SlPIP5K proteins, in which some proteins showed direct interactions, and some SlPIP5K proteins interacted with other proteins together in tomato regulation.

The expression patterns of the *SlPIP5K*s are diverse, as shown by qRT-PCR, suggesting that these genes perform a variety of physiological functions to help plants adapt to environmental challenges and that *SlPIP5K* genes are expressed in different tissues, but also show tissue expression specificity. The four *PIP5K* genes were expressed at different developmental periods in the different tissues of watermelon, but the expression of *ClaPIP5K5* was only significantly upregulated during growth, suggesting that the *ClaPIP5K* genes in watermelon may be individually or synergistically involved in regulating the morphogenesis of different tissues in the watermelon [4]. Eight *PIP5K* genes in wheat anthers respond to high-temperature stress [2], and the expression of fourteen *PIP5K* genes in tomato are differentially induced or repressed by salt and hormones. *Arabidopsis thaliana* is induced by hormone, salt, and drought stresses, and *AtPIP5K1* mRNA is rapidly induced, suggesting that it may play a role in stress tolerance [3]. In this experiment, hormone spraying during fruit development revealed that *SlPIP5K2*, *SlPIP5K3*, *SlPIP5K6*, *SlPIP5K7*, *SlPIP5K8*, *SlPIP5K9*, *SlPIP5K13*, *SlPIP5K19,* and *SlPIP5K21* are all induced by NAA, EBR, and MT hormones to varying degrees, indicating that the *SlPIP5K* genes play an important regulatory role in fruit growth and development.

## 4. Materials and Methods

### 4.1. Test Materials

The materials used in this study were processed tomato ‘M82’ samples, and seeds were provided by the Institute of Horticultural Crops, Xinjiang Academy of Agricultural Sciences. The seeds were washed and placed in Petri dishes for germination. Ninety tomato seedlings of uniform growth were selected and transferred to 1/2 modified Hollanger’s nutrient solution for incubation, and when the tomato seedlings reached the 4-leaf stage, they were treated with 200 mmol·L^−1^ NaCl solution for stress. From ten to fifteen leaves were collected at 0, 0.5, 2, 4, 8, and 12 h of treatment. Three replicates of each treatment were performed with five seedlings per replication, and the collected material was snap-frozen with liquid nitrogen; part of it was used for RNA extraction, and the other part was stored at −80 °C in a refrigerator.

The fruit variety used was a potted seedling of cherry tomato ‘Jingfan Pink Star No. 1’. The cherry tomatoes were sprayed with different concentrations of hormones, including 2,4-epibrassinolide (EBR), melatonin (MT), naphthaleneacetic acid (NAA), and a control group (Table 3) at the expansion, green ripening, color change, and red ripening stages. The fruits were already set at the time of expansion, i.e., the experimental fruit has been selected for green ripening, color change, and red ripening. By the time the fruit reached the red ripening stage, the fruit had also been cumulatively sprayed with the corresponding concentrations of exogenous NAA, EBR, and MT in the first three periods. The sprayed fruits were sampled by zone at the expansion, green ripening, color change, and red ripening stages. Three biological replicates were set up for each treatment and stored at −80 °C after quick freezing in liquid nitrogen.

### 4.2. Identification of the Tomato SlPIP5K Family

To identify the potential *PIP5K* family genes in tomatoes, this study was conducted using the tomato genome database (http://solomics.agis.org.cn/tomato/ (accessed on 10 April 2023)). The tomato Heinz 1706 genome data were downloaded, and the EnsemblPlants database (http://plants.ensembl.org/ (accessed on 11 April 2023)) was used to obtain genome data for Arabidopsis, potato, and grape. Then, from the Pfam database (http://pfam-legacy.xfam.org/ (accessed on 11 April 2023)), the hidden Markov model of the PIP5K domain (ID: PF01504) was downloaded, and the HMMER tool was used to search and compare the domain of the whole-genome protein of tomato PIP5K, E-value ≤ 1 × 10^−5^ [35]. Finally, through the NCBI (https://www.ncbi.nlm.nih.gov/ (accessed on 11 April 2023)) Conserved Domain Database program and Pfam (http://pfam-legacy.xfam.org/ (accessed on 11 April 2023)), the search program in the online software verified all the obtained proteins.

### 4.3. Physiochemical Analysis of SlPIP5K Gene Family Proteins in Tomato

Through ExPASy online software (https://web.expasy.org/tools/ (accessed on 13 April 2023)), the physicochemical properties of the tomato PIP5K protein were determined [36].

### 4.4. Structural Analysis of the SlPIP5K Gene Family in Tomato

The PIP5K protein sequences of tomato, Arabidopsis, potato, and grape were compared using Muscle in MEGA7.0 software, and the adjacent-joining (NJ) method was used to analyze and construct the family phylogenetic tree. The bootstrap repeat value was 1000, and the online website EVOLVIEW was used (https://www.evolgenius.info/evolview-v2/ (accessed on 14 April 2023)) to visually improve the evolutionary tree [37]. MEME (https://meme-suite.org/tools/meme/ (accessed on 15 April 2023)) was used to perform conservative motif analysis [38], and TBtools software (V 2.028) was used to draw the gene structure map and protein domain distribution maps [39].

### 4.5. Chromosome Location, Collinearity and Ka/Ks Analysis of the SlPIP5K Gene in Tomato

According to the chromosome position information for the *PIP5K* genes, a position distribution map was drawn using TBtools software. Among them, MCScanX was used for the collinearity analysis of the *PIP5K* genes in tomato, Arabidopsis, potato, and grape [40]; the Circos tool was used to generate a chromosome collinearity map [41]; and the Ka/Ks ratio of tomato collinear gene pairs was obtained using the Ka/Ks tool [42].

### 4.6. Cis-Element Analysis of the SlPIP5K Gene in Tomato

The sequence 2000 bp upstream of the *SlPIP5K* genes was extracted using TBtools software, and the sequence was displayed on the online site of Plant Care (http://bioinformatics.psb.ugent.be/webtools/plantcare/html/ (accessed on 16 April 2023)) Cis-acting element of the predicted promoter region [43]. Finally, the HeatMap tool in TBtools software was used to make the expression heatmap.

### 4.7. Analysis of the Expression Pattern of the SlPIP5K Gene in Tomato

Transcriptome data downloaded from the NCBI database (https://www.ncbi.nlm.nih.gov/ (accessed on 18 April 2023)) under accession numbers PRJNA19188 and PRJNA8877 were used to interpret the expression pattern of the *SlPIP5K* genes in different tissues and under salt stress, PRJNA192978 and PRJNA888477, respectively. The HeatMap module in TBtools was used to generate heatmaps of *SlPIP5K* genes’ expression patterns, and then the generated heatmaps were improved using AI software (Adobe Illustrator 2020, V 24.0.0.328).

### 4.8. Analysis of the SlPIP5K Protein Interaction Network

The protein interaction relationships were predicted with the STRING online website (https://string-db.org/ (accessed on 20 April 2023)), and the interaction networks were displayed using Cytoscape software (V 3.9.1) [44,45].

### 4.9. Real-Time Fluorescence Quantitative PCR

The total RNA was extracted using the RNA-prep Pure Plant Kit (DP441) from Tiangen Biochemical Technology (Beijing) Co. (Beijing, China) The qRT-PCR system (20 μL) contained 10 μL 2× ChamQ Universal SYBR qPCR Master Mix (Novozymes, Franklinton, NC, USA), 8.2 μL ddH_2_O, 0.4 μL each of upstream- and downstream-specific primers, and 1 μL of cDNA template. Using a Roche LightCycler 96 real-time quantitative fluorescence PCR instrument, the reaction procedure involved 120 s of predenaturation at 94 °C, 5 s of denaturation at 94 °C, 15 s of annealing time, and 10 s of extension at 72 °C for 45 cycles. The relative expression was calculated using three biological replicates and the 2^−ΔΔCT^ method using tomato Actin as the internal reference gene, and the relative expression was the relative value of the treated and control groups [46]. Fluorescent quantitative PCR primers were designed based on the conserved sequences of tomato *SlPIP5K* gene family members (Table 4).

### 4.10. Subcellular Localization of the Tomato SlPIP5K2 Gene

To study the transient expression of the PIP5K protein, it was subcloned and inserted into an EGFP-localized vector, which was then detected by PCR and verified by sequencing, after which, the recombinant vector was transformed by *Agrobacterium tumefaciens* (GV3101)-mediated methods, and then injected into tobacco leaves for 12 h in a dark culture. Finally, after 3 d of normal incubation, the infested area was cut from the leaves of the tobacco seedlings, and the epidermal sections were torn and placed under a laser confocal microscope to observe the fluorescence phenomenon.

## 5. Conclusions

In this study, we used bioinformatics to screen and identify the tomato *PIP5K* gene family to study the structure and conserved domains of these genes and to analyze their properties in terms of phylogeny, chromosomal distribution, protein physicochemical properties, and motif prediction. At the same time, the family member gene promoter cis-acting elements, expression patterns, and protein interaction networks were analyzed. In addition, the expression of the *SlPIP5K* genes was analyzed based on RNA-seq and qRT-PCR in different tissues of tomato as well as during salt stress and hormone treatments, revealing the genes involved in the response to salt stress and the growth and development of the fruits, further confirming that *PIP5Ks* may play important roles in tomato salt tolerance and fruit development. This will provide a theoretical basis for the in-depth study of tomato *PIP5K* family members during abiotic stress and fruit development, as well as a reference for the selection of new tomato varieties and germplasm innovation.

## Figures and Tables

**Figure 1 ijms-25-00159-f001:**
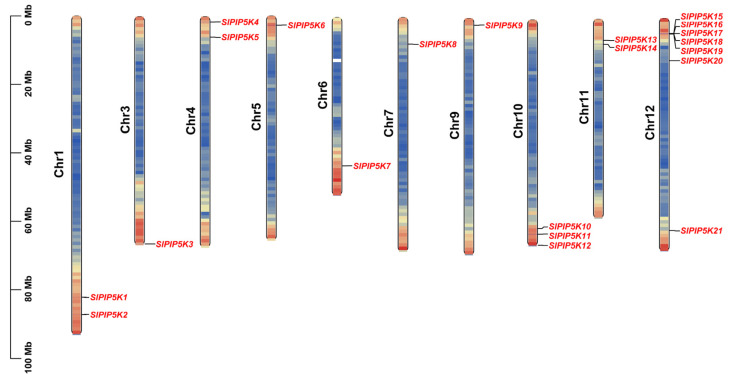
Chromosomal localization of the tomato *SlPIP5K* gene family. The names of the chromosomes are shown on the left of each chromosome. The black line indicates the gene location. The five colors on the chromosome, dark blue, light blue, yellow, light red, and dark red, represent the gene density, with red representing high levels and blue representing low levels. The size of the chromosome is listed in metabases (Mb).

**Figure 2 ijms-25-00159-f002:**
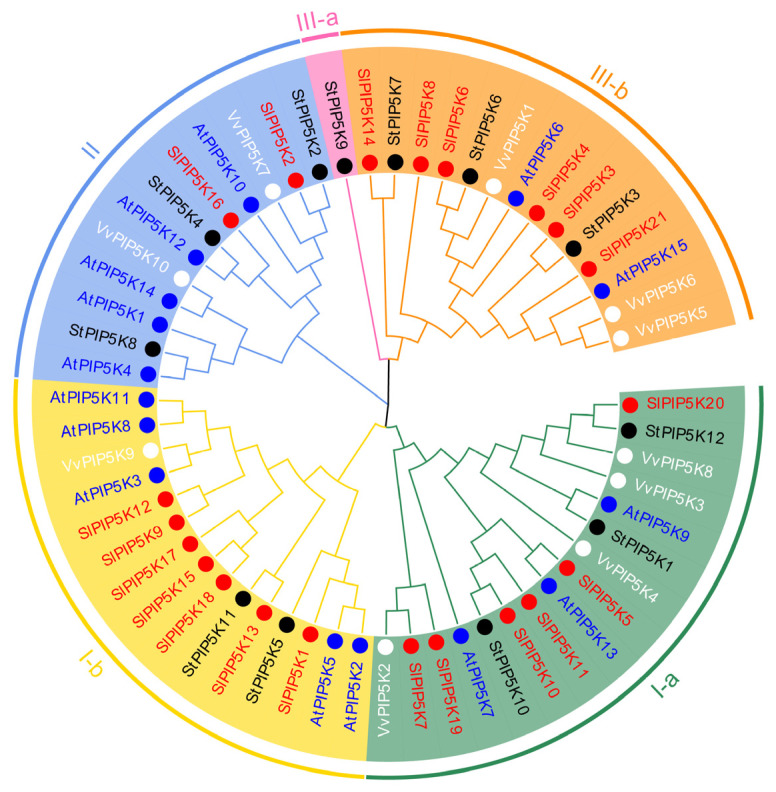
Phylogenic tree for the *PIP5K* gene family members from *Solanum lycopersicum* (Sl), *Arabidopsis thaliana* (At), *Solanum tuberosum* (St), and *Vitis vinifera* (Vv). Different colored words represent different plants, and different clades are represented by different colors.

**Figure 3 ijms-25-00159-f003:**
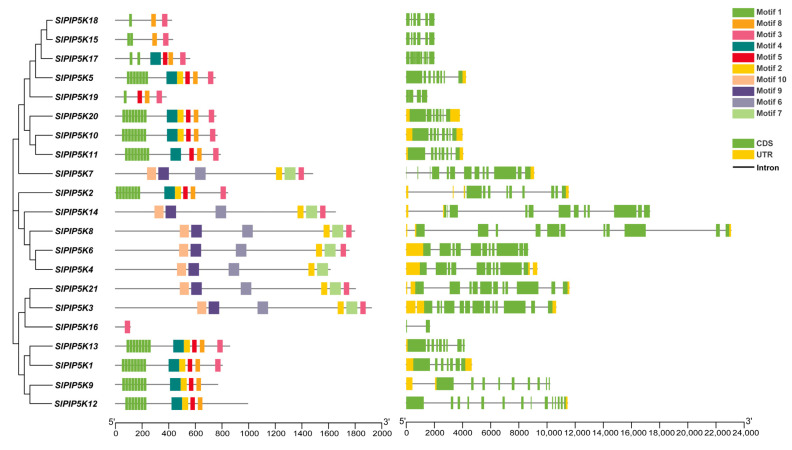
The gene structure and motif of the *SlPIP5K* transcription factor in tomato.

**Figure 4 ijms-25-00159-f004:**
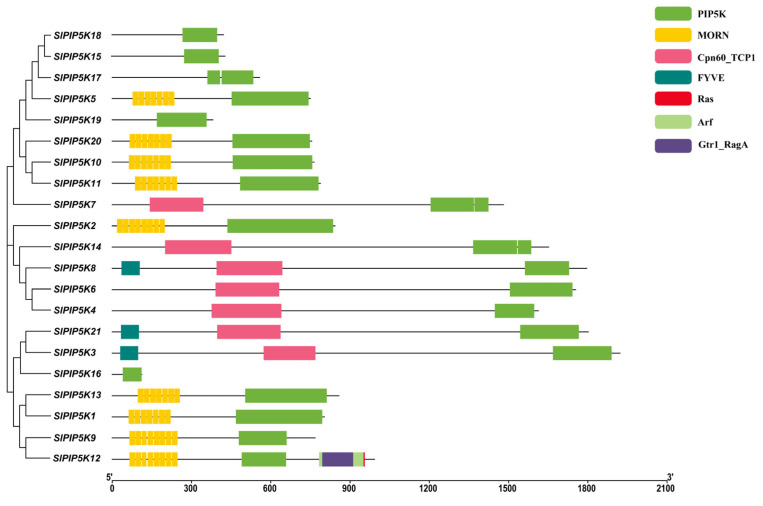
Protein domain distribution of *SlPIP5K* family members in tomato.

**Figure 5 ijms-25-00159-f005:**
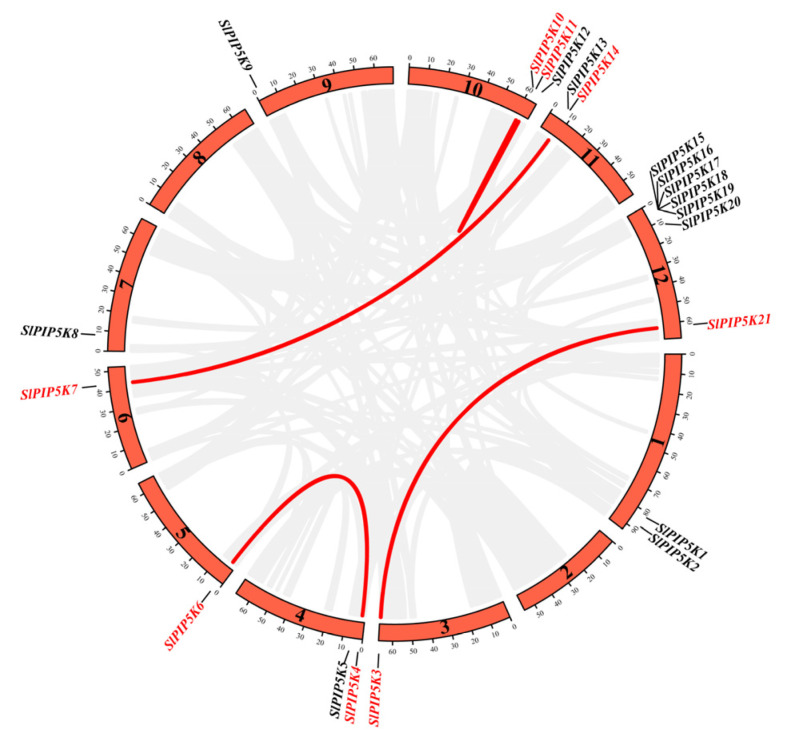
Intraspecies collinearity analysis of the *SlPIP5K* family genes in tomato. The gray lines in the background show all the syntenic blocks in the tomato genome, and the red lines show the syntenic gene pairs of *SlPIP5K* genes.

**Figure 6 ijms-25-00159-f006:**
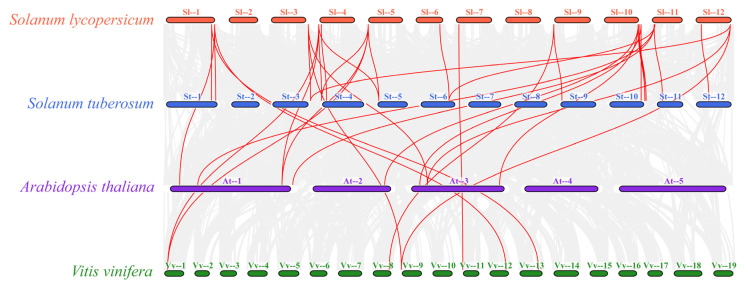
Syntenic relationship of *PIP5K* genes in different species. The gray lines in the background show the collinear blocks within tomato and other plant genomes, while the red lines highlight the syntenic gene pairs of *SlPIP5K* genes.

**Figure 7 ijms-25-00159-f007:**
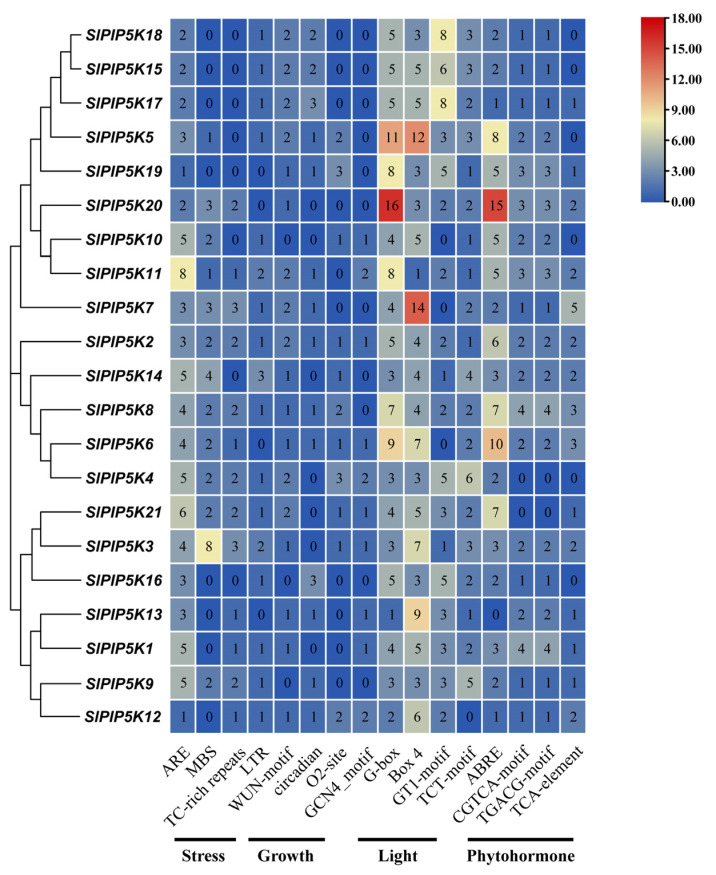
Cis-element distribution of tomato *PIP5K* genes. Dark blue, light blue, yellow, light red, and dark red are used to represent gene expression levels. Blue indicates weak gene expression, and red indicates strong gene expression.

**Figure 8 ijms-25-00159-f008:**
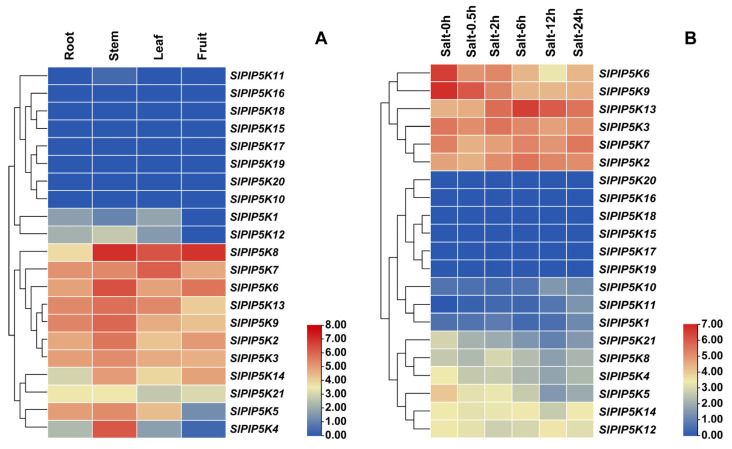
Heatmap of *SlPIP5K* gene family members using the transcriptional datasets (**A**) Heatmap of *SlPIP5K* genes expression in tomato plant tissue. (**B**) Heatmap of *SlPIP5K* genes expression in tomato leaves under salt stress. Dark blue, light blue, yellow, light red, and dark red are used to represent gene expression levels. Blue indicates weak gene expression, and red indicates strong gene expression.

**Figure 9 ijms-25-00159-f009:**
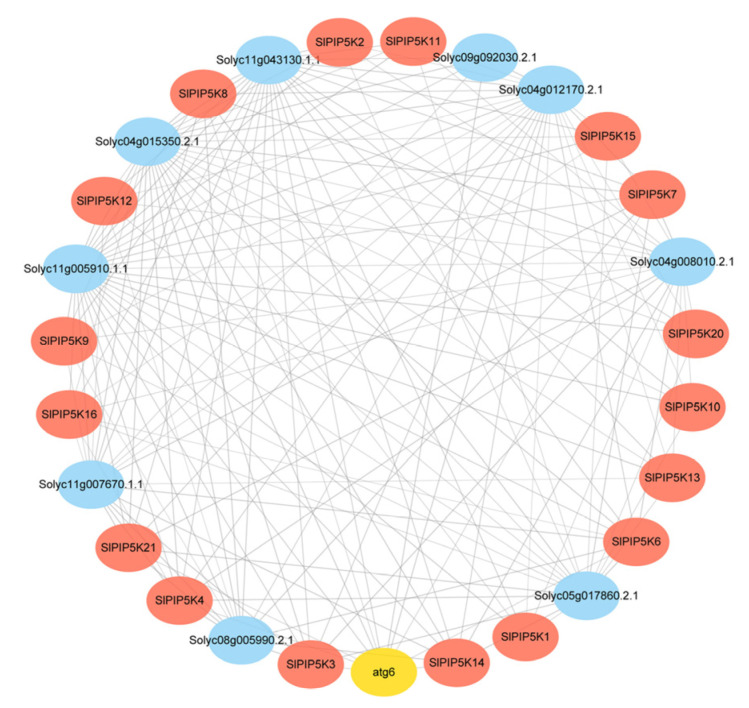
Intercrossing network diagram of *PIP5K* genes in tomato.

**Figure 10 ijms-25-00159-f010:**
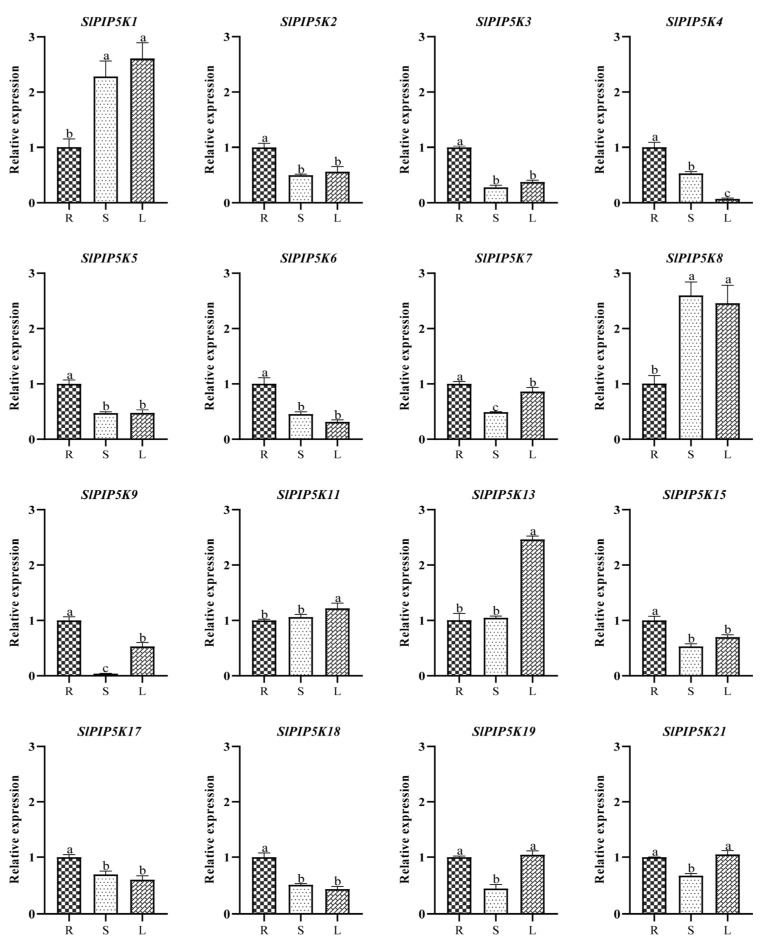
Expression of *SlPIP5K* genes in different tissues of tomato. R: root; S: stem; L: leaf. Different lower-case letters indicate significant differences between means as measured by ANOVA followed by Duncan’s multiple range test (*p* < 0.05).

**Figure 11 ijms-25-00159-f011:**
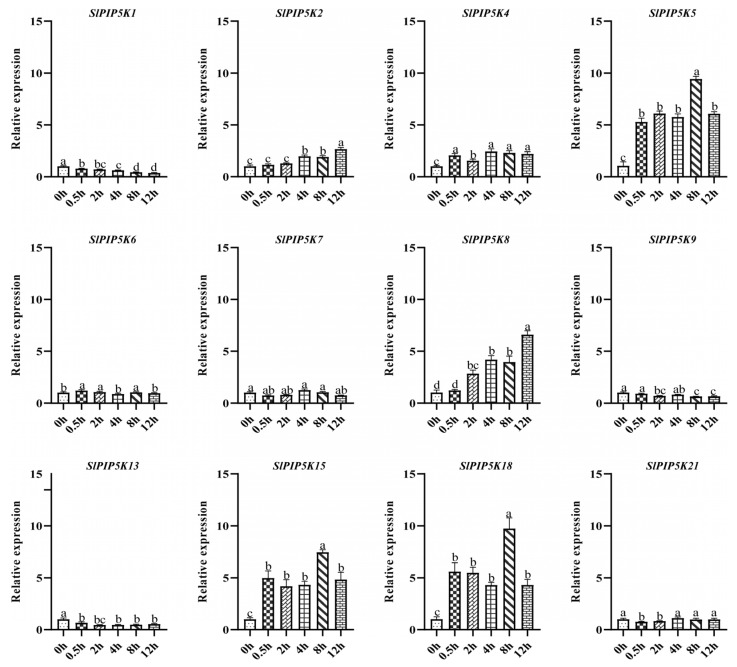
Expression of tomato *SlPIP5K* genes under salt stress. Different lower-case letters indicate significant differences between means as measured by ANOVA followed by Duncan’s multiple range test (*p* < 0.05).

**Figure 12 ijms-25-00159-f012:**
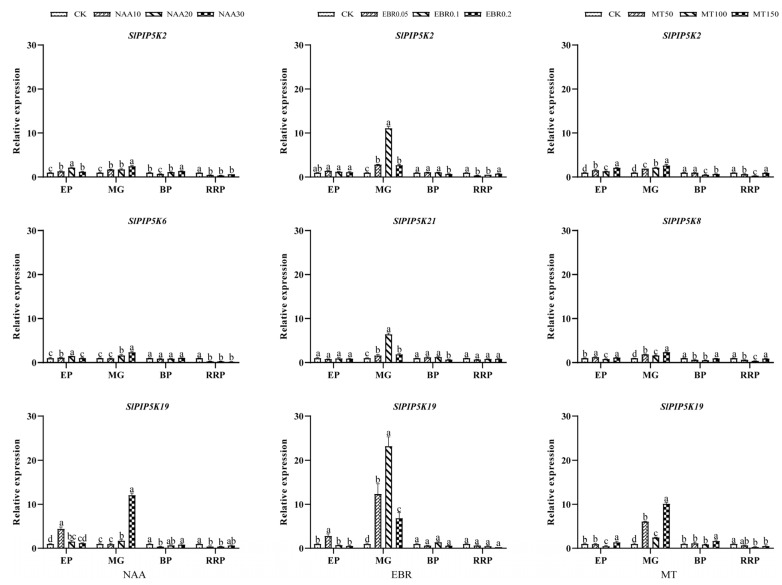
Expression of tomato *SlPIP5K* genes under different hormones. EP: swelling stage; MG: green ripening stage; BP: color conversion stage; RRP: red ripening stage. Different lower-case letters indicate significant differences between means as measured by ANOVA followed by Duncan’s multiple range test (*p* < 0.05).

**Figure 13 ijms-25-00159-f013:**
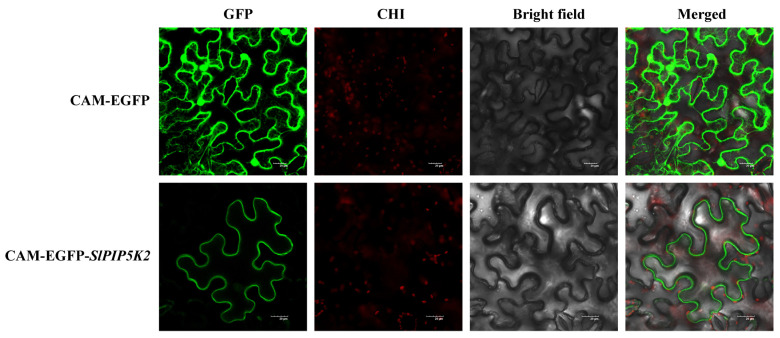
Subcellular localization of *SlPIP5K2*. GFP: green fluorescent protein; CHI: chloroplast fluorescence channel; bright field: visible light; merged: overlay of bright field, green fluorescence, and red fluorescence images. Bar = 20 μm.

**Table 1 ijms-25-00159-t001:** Physicochemical properties of amino acid sequences encoded by members of the tomato *SlPIP5K* gene family.

Gene Name	Gene ID	Chromosome Location	Amino Acid	Molecular Weight	Isoelectric Point	Gravy	Protein Secondary Structure		
							a	b	c
*SlPIP5K1*	Solyc01T003005.1	chr1	802	90,622.79	8.61	−0.647	26.18	8.73	47.76
*SlPIP5K2*	Solyc01T003618.1	chr1	842	95,980.04	9.01	−0.377	26.13	8.08	46.91
*SlPIP5K3*	Solyc03T003552.1	chr3	1921	214,107.32	5.53	−0.341	31.34	3.85	50.55
*SlPIP5K4*	Solyc04T000183.1	chr4	1612	180,971.71	6.07	−0.465	32.75	3.41	50.06
*SlPIP5K5*	Solyc04T000631.1	chr4	749	84,051.01	5.70	−0.638	22.96	9.08	49.80
*SlPIP5K6*	Solyc05T000316.1	chr5	1753	196,588.79	5.71	−0.463	33.31	4.28	48.20
*SlPIP5K7*	Solyc06T001582.1	chr6	1480	165,198.92	5.25	−0.443	33.65	3.92	49.73
*SlPIP5K8*	Solyc07T000618.2	chr7	1795	200,276.95	6.08	−0.434	31.59	3.84	50.97
*SlPIP5K9*	Solyc09T000252.1	chr9	767	86,529.11	8.59	−0.495	24.12	9.00	49.15
*SlPIP5K10*	Solyc10T002241.1	chr10	763	87,038.01	6.47	−0.639	23.98	9.31	48.49
*SlPIP5K11*	Solyc10T002463.1	chr10	787	89,822.06	5.81	−0.621	25.79	8.89	47.52
*SlPIP5K12*	Solyc10T002914.2	chr10	992	110,916.05	9.12	−0.450	24.80	8.77	46.67
*SlPIP5K13*	Solyc11T000703.1	chr11	857	97,104.71	8.54	−0.609	25.67	7.12	48.89
*SlPIP5K14*	Solyc11T000796.1	chr11	1651	184,062.73	5.42	−0.482	32.53	3.76	50.82
*SlPIP5K15*	Solyc12T000568.1	chr12	429	48,871.65	4.90	−0.647	27.97	10.72	41.03
*SlPIP5K16*	Solyc12T000573.1	chr12	111	12,389.00	6.07	−0.328	24.32	5.41	44.14
*SlPIP5K17*	Solyc12T000577.1	chr12	557	62,248.04	5.31	−0.560	25.31	9.52	45.96
*SlPIP5K18*	Solyc12T000581.1	chr12	421	47,776.37	4.80	−0.629	30.64	11.40	34.92
*SlPIP5K19*	Solyc12T000584.1	chr12	380	42,934.34	4.90	−0.485	35.79	6.84	39.21
*SlPIP5K20*	Solyc12T000991.1	chr12	754	85,902.92	8.15	−0.678	24.67	9.55	47.61
*SlPIP5K21*	Solyc12T002108.1	chr12	1801	199,991.07	5.48	−0.420	29.65	4.05	52.69

Note: a refers to α-helix, b refers to β-folding, c refers to irregularly curled.

**Table 2 ijms-25-00159-t002:** Ka/Ks Analysis of the Tomato *SlPIP5K* Gene.

Duplicate Gene Pairs	Ka	Ks	Ka/Ks	Duplicated Type
SlPIP5K10/SlPIP5K11	0.155179075	0.771386845	0.201168941	Segmental
SlPIP5K14/SlPIP5K7	0.196236825	0.653379471	0.300341277	Segmental
SlPIP5K21/SlPIP5K3	0.157812019	0.618222042	0.255267539	Segmental
SlPIP5K4/SlPIP5K16	0.152163772	0.501943081	0.303149457	Segmental

**Table 3 ijms-25-00159-t003:** Hormone concentration gradient.

Treatment	Concentration
Control	Distilled water + 0.1% Tween 80 + 0.1% Ethanol
EBR	0.05 mg·L^−1^ EBR + 0.1% Tween 80 + 0.1% Ethanol
EBR	0.1 mg·L^−1^ EBR + 0.1% Tween 80 + 0.1% Ethanol
EBR	0.2 mg·L^−1^ EBR + 0.1% Tween 80 + 0.1% Ethanol
MT	50 μmol·L^−1^ MT + 0.1% Tween 80 + 0.1% Ethanol
MT	100 μmol·L^−1^ MT + 0.1% Tween 80 + 0.1% Ethanol
MT	150 μmol·L^−1^ MT + 0.1% Tween 80 + 0.1% Ethanol
NAA	10 mg·L^−1^ + 0.1% Tween 80 + 0.1% Ethanol
NAA	20 mg·L^−1^ + 0.1% Tween 80 + 0.1% Ethanol
NAA	30 mg·L^−1^ + 0.1% Tween 80 + 0.1% Ethanol

**Table 4 ijms-25-00159-t004:** Primer sequences for qRT-PCR experiments.

Gene	Forward Primer Sequence (5′ → 3′)	Reverse Primer Sequence (5′ → 3′)
*SlPIP5K1*	GCCAGATGGACAAGGGAGATA	CGTGCCATTCCCTTTAGGA
*SlPIP5K2*	CATCACTGAAATGGTCCTCTCC	CAAACCGTCCTTCTCCAGACA
*SlPIP5K3*	TGGGTGTGTTAGAGTCTCCTGG	CATCTTGGACATCGTTGGCT
*SlPIP5K4*	TCCAGTATGCCGTCTTTGCG	TATGGTGTCTTGCCGAGGA
*SlPIP5K5*	TCGCATTACGCCGACGAAT	CTTTACCACTCGCCTTCCCTCT
*SlPIP5K6*	TTGGGCATTCTGTTCAGGAC	TGGTCTCGGTAAACGGACA
*SlPIP5K7*	ATCGCCTGGAGAGAGTTGCT	ACAGCCCTCAATGAACAACAG
*SlPIP5K8*	CCAGAGCCAGAAACAGAAGAGG	ATTGCCTTCCTATGCTCCG
*SlPIP5K9*	TGGAGGATGCTATGAGTGCTC	ATGATTGCTTCACCTGGTCTCT
*SlPIP5K11*	GCAGTTGGGAATCAGGCATAC	GAACTCACAAGACTGATGGCG
*SlPIP5K13*	GAAGAAGAGGTCATCCGTGGA	AACCCAAAGCCATCCCTGT
*SlPIP5K15*	TGCGTGAAACTGTGAAGAAACC	CATCATTCGTCGGCGTTATG
*SlPIP5K17*	TGCGTGAAACTGTGAAGAAACC	ATCATCATTCGTCGGCGT
*SlPIP5K18*	TGCGTGAAACTGTGAAGAAACC	CATCATTCGTCGGCGTTATG
*SlPIP5K19*	TGGTGGTGGTAATGGTGGTC	TCCTCCTCCTCCTCATTCCTA
*SlPIP5K21*	AGATGCCTGAGATGTCCACG	TGTAACGAATGCCCGCAA
*Actin*	CAGGGTGTTCTTCAGGAGCAA	GGTGTTATGGTCGGAATGGG

## Data Availability

Data are contained within the article and Appendix A.

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
