# Peer review of "Identification and Analysis of the Expression of the PIP5K Gene Family in Tomatoes"

_ijms, 2023, doi:10.3390/ijms25010159_

Round 1

Reviewer 1 Report

Comments and Suggestions for Authors

Dear Authors:

This study raises very interesting topics.

Congratulations on your study on molecular and cellular, physiological, and biochemical perspectives are needed to cope with and adapt to environmental stresses.

The list of specific comments is given bellow:

1.       Is this research related to your earlier study: "Genome-Wide Identification and Expression Analysis of the PLATZ Transcription Factor in Tomato", which, as you quote: "This study will ... also provide a theoretical basis for the selection and breeding of new tomato varieties and germplasm innovation”[July 2023Plants 12(14):2632;  DOI: 10.3390/plants12142632],

2.       I understand that the research is aimed at increasing tomato yields, and help in its better production, but you don't write anything, why increase them and what importance do tomatoes have for human health?,

3.       You write: „ (…) and the production of tomato in our country accounts for approximately 1/4 of the world's production of tomato; therefore, the discovery of the potential function of genes within the tomato will provide important theoretical support for the enhancement of the 66 yield and quality of tomato, as well as resistance to adversity stresses (…)”, but you didn't explain what health benefits tomatoes have and why interfere with their genome, or just to increase yields? What is the consumption of tomatoes worldwide, in Europe? And isn't current production enough?

4.       Figure 3 i Figure 4 - the font is unreadable,

5.       Figure 7 - explain the color intensity (0.00-18.00) - as in Figure 8,

6.       Figure 8 - what program was it made in?

7.       In Figure 8A  - there are values 0.00-8.00, and Figure 8B – 0.00-7.00 - Shouldn't the range of values be the same?

8.       There is no reference to Figure 9 in the text,

9.       Line 243 and 244: Replace „As shown in Figure 8A, showed that the expression…” with „As shown in Figure 8A, the expression…”,

10.    Line 252: Replace „Figure 8B shows, the highest …” with „Figure 8B shows the highest …”,

11.    Figure 10 - on the Y axis (Relative expression) there should be the same values everywhere, but there are different values - 0, 1, 2, 3, 4 or 0.0, 0.5, 1.0, 1.5 - please standardize

12.    Figure 11 - similarly to Figure 10 - please standardize the values.

Best regards,

Reviewer

Comments on the Quality of English Language

Please standardize the tenses in this work and correct minor repetitions of words.

Reviewer 2 Report

Comments and Suggestions for Authors

For improving the quality of the manuscript there are some recommendations in  4. Materials and methods:  4.1. Test material:

1.How many Tomato seedlings of uniform growth were selected ?

2. How many  leaves were collected for every treatment (at 0, 0.5, 2, 4, 6, 8 and 12 h) ?

Reviewer 3 Report

Comments and Suggestions for Authors

Dear Editors,

Congratulation for your manuscript, I will share my comments below.

The Abstract is adequate, both in terms of content and form.

References in parentheses in the Introduction chapter are formally incorrect, they should not be marked as an index. The part in which it is stated in what other species the tested gene has already been detected, I think it should be moved forward. I ask the authors to reconsider and modify this.

The Results chapter is fine, the figures and tables are transparent, but please authors to make the intertextual references to the figures follow the figures more closely, because this way the chapter loses a lot of its coherence.

The Material and Methods chapter, Discussion and Conclusion chapters are fine.

References is not formally correct, so I ask the authors to fix it according to MDPI requirements.

Author Response

Dear Editors and Reviewers:

Thank you for your letter and for the reviewers’ comments concerning our manuscript entitled “Identification and expression analysis of the PIP5K gene family in tomato” (ID: ijms-2739114). Those comments are all valuable and very helpful for revising and improving our paper, as well as the important guiding significance to our researches. We have studied comments carefully and have made correction which we hope meet with approval. Revised portion are marked in red in the paper.

The main corrections in the paper and the responds to the reviewer’s comments are as flowing:

Comments 1: How many Tomato seedlings of uniform growth were selected ?

Comments 2: How many leaves were collected for every treatment (at 0, 0.5, 2, 4, 8 and 12 h) ?

Response 1 and Response 2: Thank you for pointing this out. We agree with this comment. Therefore, we have added these: Ninety tomato seedlings of uniform growth were selected and transferred to 1/2 modified Hollanger's nutrient solution for incubation, and when the tomato seedlings reached the 4-leaf stage, they were treated with 200 mmol·L-1 NaCl solution for stress. Ten to fifteen leaves were collected at 0, 0.5, 2, 4, 8 and 12 h of treatment. Three replicates of each treatment were performed with five seedlings per replication, and the collected material was snap-frozen with liquid nitrogen, part of it was used for RNA extraction, and the other part was stored at -80°C in a refrigerator.(Line 494-Line 501)

Special thanks to you for your good comments.